



# Downscaled hyper-resolution (400 m) gridded datasets of daily precipitation and temperature (2008-2019) for East Taylor subbasin (western United States)

Utkarsh Mital[1], Dipankar Dwivedi[1], James B. Brown[2], Carl I. Steefel[1]

[1]Energy Geosciences Division, Lawrence Berkeley National Laboratory, Berkeley, CA, 94720, USA
[2]Environmental Genomics and System Biology, Lawrence Berkeley National Laboratory, Berkeley, CA, 94720, USA

*Correspondence to*: Utkarsh Mital (umital@lbl.gov)

**Abstract.** High resolution gridded datasets of meteorological variables are needed in order to resolve fine-scale hydrological gradients in complex mountainous terrain. Across the United States, the highest available spatial resolution of gridded datasets of daily meteorological records is approximately 800 m. This work presents gridded datasets of daily precipitation and mean temperature for the East-Taylor subbasin (in western United States) covering a 12-year period (2008-2019) at a high spatial resolution (400 m). The datasets are generated using a downscaling framework that uses data-driven models to learn relationships between climate variables and topography. We observe that downscaled datasets of precipitation and mean temperature exhibit smoother spatial gradients compared to their coarser counterparts. Additionally, we also observe that when downscaled datasets are reaggregated to the original resolution (800 m), the mean residual error is almost zero, ensuring spatial consistency with the original data. Furthermore, the downscaled datasets are observed to be linearly related to elevation, which is consistent with the methodology underlying the original 800 m product. Finally, we validate the spatial patterns exhibited by downscaled datasets via an example use case that models lidar-derived estimates of snowpack. The presented dataset constitutes a valuable resource to resolve fine-sale hydrological gradients in the mountainous terrain of the East-Taylor subbasin, which is an important study area in the context of water security for southwestern United States and Mexico. The dataset is publicly available at https://doi.org/10.15485/1822259 (Mital et al., 2021).

## 1 Introduction

Water resources are under increasing stresses due to Earth System change and increasing demand for clean water, food, and energy (Vörösmarty et al., 2010). The stresses on water availability and quality are felt through watersheds as they are the fundamental functional units of the Earth's surface that integrate the effects of vegetation, fluvial systems, soils and subsurface on water resources (National Research Council, 1999). Sustainable management of water resources, therefore, requires quantitative modeling efforts at the river basin scale. Such efforts involve the use of land-surface and ecohydrological models which need access to climate forcing via gridded datasets of meteorological variables. Across the United States, the highest available spatial resolution of gridded datasets of daily meteorological records is approximately



800 m (Daly et al., 2008) to 1 km (Thornton et al., 2020). This resolution does not allow models to resolve fine-scale gradients of hydro-biogeochemical processes which introduces uncertainty associated with predicting response of water resources to various drivers such as wildfire, drought, floods, land-use change, extreme weather, sea-level rise, and climate change (e.g., Singh, 1997; Cotter et al., 2003; Beven et al., 2015). Consequently, there is a need to generate gridded datasets of meteorological variables at hyper resolutions. In this work, we define hyper resolutions as spatial resolutions that are of

the order of a few hundred meters.

It is possible to obtain hyper-resolution gridded observations of meteorological variables. For example, precipitation can be measured at a resolution of 100 m via X-band radar (e.g., Feldman et al., 2021). However, high measurement cost implies that such data have limited spatial and temporal extent. This leaves us with two possible approaches to generate hyper-resolution gridded datasets that have large spatial and temporal extents: (i) spatial interpolation of point

measurements, or (ii) spatial downscaling of existing lower resolution datasets. Spatial interpolation of point measurements requires a high density of stations for high-resolution gridding to adequately capture the climatological variability (Bierkens, 2015; Beven et al., 2015). For instance, a resolution of 1 km needs a station every 1 x 1 km (Haylock et al., 2008). Since such a high station density is not feasible, interpolation approaches typically incorporate physiographic and climatological information in their methodologies while generating gridded datasets at hyper spatial resolutions (e.g., Daly et al., 2008;

Thornton et al., 2021; Lussana et al., 2019; Crespi et al., 2021; Škrk et al., 2021). A general lack of knowledge about weather patterns at fine spatial scales combined with computational expense makes it challenging to interpolate point measurements at hyper-resolutions (Daly, 2006; Beven et al., 2015). In this work, we resort to the latter approach and generate hyper-resolution datasets by spatially downscaling existing high-resolution (800 m) datasets.

There are two broad classes of techniques for downscaling: dynamical and statistical. Dynamical downscaling

involves the use of regional climate models (RCMs) whose boundary conditions are specified using coarse-resolution outputs of a general circulation model (GCM) or reanalysis datasets. RCMs perform downscaling by accounting for the effects of complex topography, surface characteristics, land-sea contrasts and other dynamical processes (Giorgi, 2019; Tapiador et al., 2020). Although these models simulate physical processes, they are computationally intensive which limits the resolution of downscaled data to a few kilometers at best (Giorgi, 2019; Tapiador et al., 2020). This has motivated the

development of statistical downscaling approaches, where a statistical or empirical relationship is modeled between high-resolution predictors and low-resolution climate variables to generate high-resolution climate data. Statistical approaches are flexible and enable downscaling of coarse-resolution data (from GCMs and reanalysis datasets) to spatial scales of individual weather stations  (e.g., Coulibaly et al., 2005; Bürger et al., 2012; Sachindra et al., 2018; Vandal et al., 2019; Gutiérrez et al., 2019; Nourani et al., 2019). The ability of statistical downscaling to generate data at such fine scales motivates us to leverage

its potential for generating gridded datasets at hyper-resolutions.

A number of recent studies have applied machine learning techniques to statistical downscaling. Machine learning techniques have the benefit of not needing to specify a functional relationship between low-resolution and high-resolution data. Several studies have sought to exploit the temporal dependencies among predictor variables by using temporal neural



networks (e.g., Coulibaly et al., 2005; Mouatadid et al., 2017; Misra et al., 2018). Other studies have performed statistical

downscaling by exploiting the spatial dependencies between low-resolution and high-resolution data (Vandal et al., 2017; Liu et al., 2020; Baño-Medina et al., 2020). Studies have also been conducted with the objective of comparing the performance of different machine learning techniques for statistical downscaling (Sachindra et al., 2018; Vandal et al., 2019).

A key challenge associated with machine learning techniques is the need for paired low-resolution and high-

resolution data for training the downscaling model. This makes it difficult to downscale data to hyper-resolutions (i.e., few hundred meters) where a ground-truth is not available. Recently, Groenke et al. (2020) presented a machine learning framework based on unsupervised, generative downscaling. However, the viability of their method was evaluated at relatively coarse resolutions (~12.5 km). Existing approaches for downscaling climate variables to hyper-resolutions specify simple functional forms that depend on elevation and nearby point observations (Fiddes and Gruber, 2014; Sen Gupta and

Tarboton, 2016; Rouf et al., 2020). The coefficients for these functional forms are determined using prior empirical studies (e.g., Liston and Elder, 2006; Kunkel, 1989). A downside of these functional forms is that the prescribed coefficients are defined to vary seasonally only – geographical variations need to be manually prescribed by the user. Additionally, such functional forms do not account for physiographic variations between a given grid point and nearby point observations. Specifically, observations whose locations have greater physiographic similarity with a given grid point need to be given

greater weights (Daly et al., 2008; Thornton et al., 2021). As a result, there is a scarcity of publicly available gridded meteorological datasets at hyper-resolutions.

In this work, we present hyper-resolution (400 m) gridded datasets of daily precipitation and mean temperature for the East-Taylor subbasin (in western United States) covering a 12-year period (2008-2019) at a high spatial resolution (400 m). The datasets are generated by spatially downscaling daily gridded datasets developed by the Parameter-elevation

Relationships on Independent Slopes Model (PRISM; Daly et al., 2008), available at a resolution of 800 m. The downscaling methodology comprises of a data-driven framework that does not need paired coarse-resolution and fine-resolution training data. Instead, we learn relationships between topographic features and daily climate variables (specifically, precipitation and mean temperature). The methodology also uses nearest-neighbor maps of weather stations to help constrain the learnt relationships between topography and climate variables. Subsequently, using hyper-resolution information about the

topography, the learnt relationships are used to model precipitation and mean temperature at hyper-resolution. This approach has the benefit of leveraging expert knowledge about physiographic factors and climatological processes that is embedded in the gridded datasets. We conduct an exploratory analysis of the downscaled datasets and quantify (i) how their spatial gradients vary when compared with datasets at the original resolution, (ii) the mean residual error with respect to datasets at the original resolution, and (iii) the effect of elevation on their spatial variation. Downscaled datasets provide a more precise

definition of local gradients compared to their coarse-resolution counterparts. Such a definition is beneficial, especially for ecohydrological modeling in complex mountainous terrains where gradients can occur at fine spatial scales (Crespi et al.,

2021). We observe the benefits of using downscaled datasets via an example use case that models lidar-derived estimates of snowpack.

The rest of the paper is organized as follows. Section 2 describes the study area and the various data sources.
Section 3 describes the downscaling methodology for downscaling gridded datasets, and is followed by a summary of statistical descriptors used to describe the downscaled datasets (Sect. 4). This is followed by the results (Sect. 5) and an example use case of downscaled datasets (Sect. 6). Finally, we discuss some caveats and conclusions.

## 2    Study area and data

### 2.1    Study area

Our study area is the East-Taylor subbasin (hydrologic unit code 14020001; Fig. 1), which is a mountainous watershed in Colorado, western United States. East-Taylor subbasin is an area of intensive research activity as it encompasses the East River watershed, which has a highly instrumented testbed developed for understanding the impact of watershed changes on water availability and quality (Hubbard et al., 2018). East-Taylor subbasin is also part of the Upper Colorado River Basin (UCRB; hydrologic unit code 14), which is a site of the United States Geological Survey (USGS) Next Generation Water
Observation System (NGWOS; Gallaudet and Petty, 2018). In general, UCRB is an important study area as it drains into the Colorado River which is the principal source of water and jobs for 40 million people in the southwestern United States and Mexico (James et al., 2014). The Colorado River is under increasing stress due to drought and changing seasonality of snowmelt (Milly and Dunne, 2020) which can significantly impact the regional economies. Generating hyper-resolution datasets of gridded meteorological forcing in UCRB can help with quantitative modeling efforts geared towards water
security for the region.

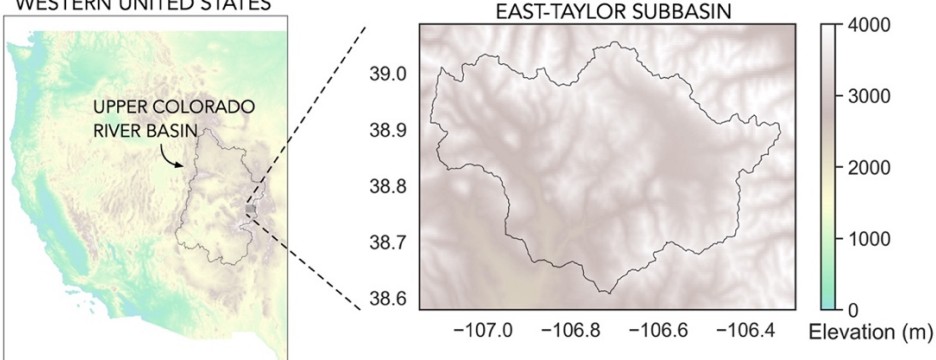

**Figure 1. Location of East-Taylor subbasin in the western United States**



### 2.2    Data sources

#### 2.2.1    Gridded meteorological data

We obtained gridded estimates of daily precipitation, maximum daily temperature, and minimum daily temperature from PRISM. The maximum and minimum values of temperature were averaged to obtain mean values of temperature. PRISM data at 800 m spatial resolution is a proprietary dataset purchased from the PRISM Climate Group at Oregon State University (https://prism.oregonstate.edu, created 3 August 2020). PRISM serves as the official spatial climate dataset of the United States Department of Agriculture (Daly et al., 2008). Its methodology (to account for orographic effects) and
climatology has been leveraged to generate various gridded datasets (Livneh et al., 2013; Abatzoglou, 2013; Behnke et al., 2016; Xie et al., 2007; Xia et al., 2012).

#### 2.2.2    Weather station data

We obtained daily weather station data from the Global Historical Climatology Network (GHCN; Menne et al., 2012). The GHCN-Daily dataset integrates daily climate observations from 80,000 stations worldwide and subjects them to a suite of
quality assurance measures.

#### 2.2.3    Elevation data

We obtained elevation maps from the National Elevation Dataset (NED; U.S. Geological Survey, 2019; https://apps.nationalmap.gov) at a spatial resolution of 10 m.

#### 2.2.4    Lidar observations of snowpack

We obtained Lidar maps of snow water equivalent (SWE) generated by the Airborne Snow Observatory (ASO) at a spatial resolution of 50 m (Painter, 2018). These maps constitute independent datasets that are used exclusively to demonstrate an example use case of the downscaled datasets, and are not used in the downscaling methodology itself. Across the East Taylor subbasin, the ASO data quantify SWE across Crested Butte (CB), Gunnison-East River (GE), and Gunnison-Taylor River (GT) basins (Fig. 2). There are eight maps across 2016 to 2019, out of which five correspond to the early-melt period
(March/April) and three correspond to the late melt period (May/June).



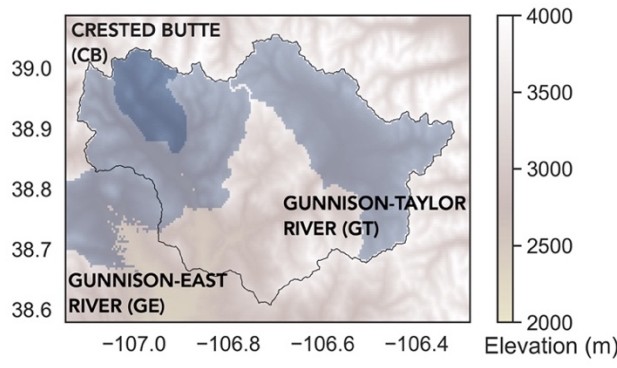

**(a)**

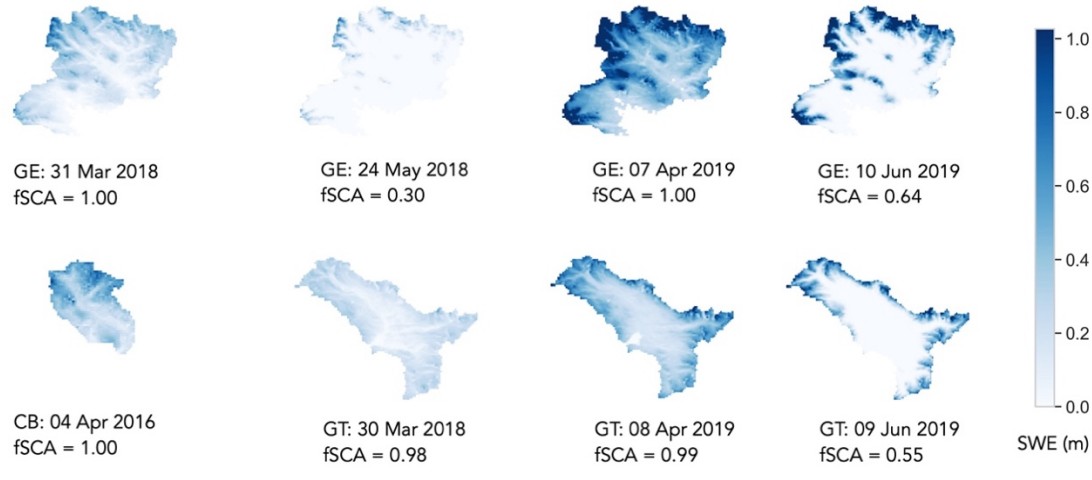

**(b)**

**Figure 2: Lidar-derived SWE maps within East-Taylor subbasin obtained via ASO; (a) spatial extent of the maps in East-Taylor, (b) actual maps (upscaled to 400 m resolution). Each map is labeled by its basin, date of acquisition, and fraction of snow-covered area (fSCA) at its native 50 m resolution.**

## 2.3 Data pre-processing

The above data streams (Sect. 2.2) were subjected to various pre-processing steps as described below.

### 2.3.1 Gridded meteorological data (PRISM)

PRISM defines a day as the 24-hour period ending at noon UTC (Strachan and Daly, 2017). The time zone of our study area is UTC-07:00 (or UTC-06:00 during daylight savings) which means that, for a given day, the 24-hour period ends at 5:00am local time (or 6:00am during daylight savings). We shifted the dates of the precipitation data backward by one day, so that the 24-hour period starts (rather than ends) at 5:00am local time. A similar adjustment was made for dates of maximum daily

temperature prior to computing values of mean daily temperature. The dates of minimum daily temperature were not
changed since the minimum temperature is likely to occur early in the morning around or before 05:00am local time.

### 2.3.2    Weather station data

Weather station data were subjected to two steps of pre-processing. The first step addresses inconsistent reporting times.
Some stations are automated and report observations that represent the 24-hour period ending at midnight. However, most
stations typically report daily observations at morning local time. The dates of precipitation and maximum temperature for
the latter group of stations were shifted backward by one day, along the lines described for PRISM data above. Thornton et
al. (2021) discuss this issue in more detail. The second step addresses gap-filling of missing values, which can happen for
various reasons, such as due to equipment malfunction, network interruptions, and natural hazards. We gap-filled missing
values of precipitation and mean temperature using a data-driven sequential imputation approach (Mital et al., 2020;
Dwivedi et al., 2022).

### 2.3.3    Elevation data

Elevation maps were upscaled using bilinear interpolation (in increments of 2x), mosaiced and reprojected to align their
grids with the PRISM data. This was done with the help of Python's *Rasterio* module (Gillies and others, 2013). The gridded
elevation data were also used to derive gridded estimates of slope and aspect using Python's *RichDEM* module (Barnes,
2016).

### 2.3.4    Lidar observations of snowpack

The lidar maps were upscaled using bilinear interpolation (in maximum increments of 2x), and reprojected to align their
grids with the PRISM data. The maps corresponding to Gunnison-East River (GE) required additional quality control
measures (see Appendix A).

## 3    Downscaling methodology

### 3.1    Data-driven model: Random Forests

We employ Random Forests (RF) to implement our spatial downscaling methodology. RF are a non-parametric machine
learning method based on an ensemble of decision trees (Breiman, 2001). Decision trees seek to minimize the error in
modeling the target variable by recursively partitioning the input feature (or predictor) space into smaller subspaces. For
regression models, a typical error criterion is the mean-squared error. The RF model employs bootstrapping to generate a
different set of data points for each decision tree. The final model output is obtained by mean aggregating the output of all
decision trees in the ensemble. RF models also provide measures of the relative "feature importance" of each predictor



variable. We implemented RF using Python's *scikit-learn* module (Pedregosa et al., 2011). The hyperparameter values employed in our RF models are specified in Appendix B.

### 3.2    Extracting relationships between topography, weather stations, and climate variables

Our data-driven downscaling methodology consists of two steps: (i) learn the mapping between topographic features and the daily climate variable (at the native resolution of 800 m), and (ii) apply the learnt mapping to model the downscaled climate variable using topographic features (at a resolution of 400 m). Figure 3 shows the schematic of the methodology. The gridded climate variable $V$ can be expressed as the following function $f$:

$$V = f(x, y, z, w_{1-10})$$    (1)


where $x, y$, and $z$ correspond to longitude, latitude, and elevation, respectively. These three spatial coordinates quantify the three-dimensional topography and enable the data-driven model to learn local relationships between $V$ and physiography – relationships that correspond to expert knowledge embedded in the PRISM dataset. Finally, $w_{1-10}$ corresponds to the ten most correlated weather stations for each grid point. We use $w_{1-10}$ as a shorthand notation for $w_1, w_2, ..., w_{10}$, where $w_i$

corresponds to the $i$-th most correlated weather station for a grid point. $w_{1-10}$ can be thought of as ten nearest neighbors for each grid point. The PRISM dataset is developed using a weighted linear regression of weather station data. By using $w_{1-10}$, we strive to give our machine learning model the same raw data that is used by the PRISM methodology. We picked ten stations since it seems to correspond to the upper limit of the minimum number of stations that PRISM uses to develop a climate-elevation regression for a grid point (Daly et al., 1994). The use of $w_{1-10}$ helps to constrain or regularize the

relationship learnt between the climate variable and topography, since it more explicitly forces the machine learning model to consider point measurement data. We determined $w_{1-10}$ using the entire 12-year time series of the climate variable and weather stations. Figure 4 shows a map of $w_1$. Note that since we are doing spatial downscaling, we learn the function $f$ separately for each day. This also enabled us to get estimates of relative feature importance of each predictor variable, which are presented in Appendix C.


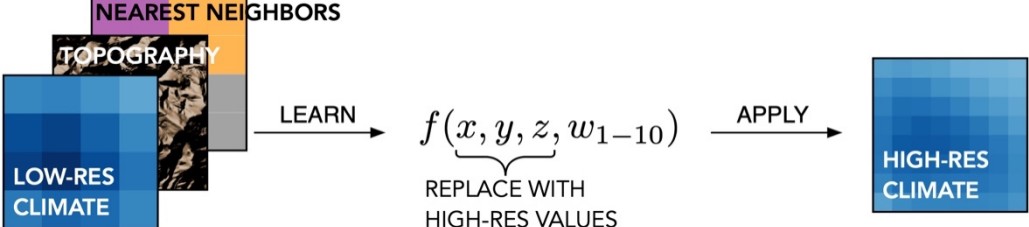

**Figure 3: Schematic of the spatial downscaling methodology**





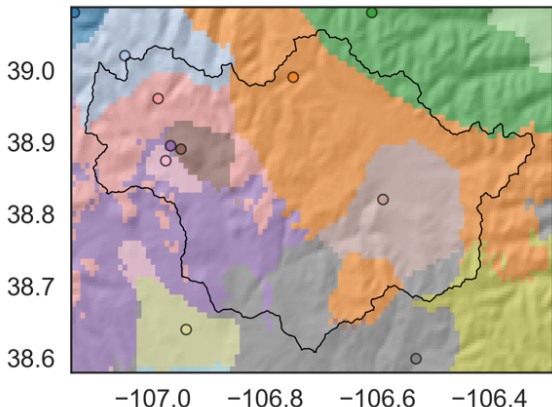

**Figure 4: Nearest neighbor map ($w_1$) of precipitation in East Taylor subbasin. Open circles correspond to locations of weather stations. Each grid point is color-coded using the color of its "nearest neighbor" weather station. Note that the nearest neighbor for a grid point is not necessarily the closest station, but the most correlated station. Some stations are outside the spatial extent of the map (not shown).**

Finally, we subjected the downscaled precipitation grids to a variable filter as described by Daly et al. (2008). The filter performs a distance-weighted average of all surrounding grid cells and ensures a smooth precipitation field in low-gradient areas, without affecting the high-gradient areas.

## 4    Exploratory analysis of downscaled and original datasets

We consider the following statistical measures (Sect. 4.1-4.3) to describe the downscaled datasets. Each measure helps focus on a salient feature of the datasets.

### 4.1    Quantifying roughness

Gridded estimates of precipitation and temperature should exhibit a spatial variation that is consistent with the resolution of the grid. Projecting coarse-resolution meteorological variables on a fine-resolution grid results in discontinuous spatial gradients which can impact the modeling of land surface processes (Maina et al., 2020). This implies that spatial gradients of the downscaled climate variables should exhibit a more gradual (or smoother) variation, when compared to their coarse-resolution counterparts. We explore the relative smoothness of the downscaled and original datasets by quantifying their roughness.

To quantify the roughness of a gridded climate variable, we start by computing its Laplacian. The Laplacian $\mathcal{L}$ of a gridded variable is estimated by convolving the gridded variable with the following 3x3 filter:



$$\begin{bmatrix} 0 & 1 & 0 \\ 1 & -4 & 1 \\ 0 & 1 & 0 \end{bmatrix}$$


The Laplacian can be used to visualize changes in gradients (or roughness) of the gridded variables. Its ability to capture the roughness of an image has long been utilized in computer vision for edge detection (Torre and Poggio, 1986). We estimate the roughness $\mathcal{J}$ of a gridded variable by summing the element-wise squares of its Laplacian:

$$\mathcal{J} = \sum_{i,j} \mathcal{L}_{i,j}$$

Where $\mathcal{L}_{i,j}$ corresponds to the $(i,j)$-th element of $\mathcal{L}$. This approach to estimate roughness is motivated by the definition of roughness penalty used for fitting smooth splines to data (Gu, 2011). We expect the gridded variable at the downscaled resolution to have a lower roughness when compared to the original resolution. To compare the roughness at the two resolutions, we define a quantity called roughness ratio ($RR$) as follows:

$$RR = \mathcal{J}_{400}/\mathcal{J}_{800} \qquad (2)$$


where the subscript corresponds to the spatial resolution of the gridded variable. $RR < 1$ implies that $\mathcal{J}_{400} < \mathcal{J}_{800}$ which means that the datasets are smoother at the downscaled resolution.

## 4.2 Quantifying residual error

If the downscaled datasets are aggregated to the original resolution, we should get back the original dataset. The deviation
between the aggregated and the original datasets (both at a resolution of 800 m) can be quantified using the mean residual error $R$:

$$R = \frac{1}{n} \sum_{j} \overline{V_j} - V_j \qquad (3)$$

where $V$ refers to the value of the climate variable in the original dataset, $\overline{V}$ refers to the aggregated value of the climate
variable obtained from the downscaled dataset, the subscript $j$ is the index of the grid point at the original resolution, and $n$ is the total number of grid points in the dataset at the original resolution. Ideally, the mean residual error should be close to zero which implies that the downscaled datasets are spatially consistent with the datasets at the original resolution.





### 4.3 Quantifying influence of elevation

PRISM assumes that for a localized region, elevation is the most important factor in the distribution of temperature and
precipitation (Daly et al., 2008). Therefore, it is of interest to investigate how elevation influences the downscaled datasets.
We do this using partial dependence plots (PDP; Friedman, 2001), which show the marginal effect that a feature (here,
elevation $z$) has on the outcome of a model. For regression, the partial dependence function is defined as (Molnar, 2019):

$$\bar{f}(z) = E_{X_C}\left[\bar{f}(z, X_C)\right] \tag{4}$$

where $z$ is the feature for which we obtain a PDP and $X_C$ are the other features used in the machine learning model $\bar{f}$. Here,
$\bar{f}$ approximates the model $f$, as defined in Eq. (1), which is used to generate the downscaled datasets. The partial
dependence function can be approximated using a Monte Carlo simulation whereby we consider all the instances of our data
(which are used to learn $\bar{f}$), and replace the true value of $z$ with a realization of $z$ instead. We can then obtain the average
model prediction for each realization of $z$. We estimated the partial dependence functions for models used to generate
downscale estimates of both precipitation and mean temperature.

## 5 Results

While computing statistical estimates, we randomly sampled 100 time points (or days). This yields a sample size that is
tractable and amenable to analysis and visualization while being large enough to yield representative estimates. For
precipitation, we considered only the wet days (when mean precipitation across East Taylor was greater than 1 mm). Dry
days imply absence of precipitation which preclude meaningful analysis. No such constraints on selection of days are needed
for analyzing mean temperature.

### 5.1 Downscaled datasets exhibit smoother spatial gradients than their coarser counterparts

Figures 5 and 6 show examples of downscaled precipitation and mean temperature fields, along with their original
counterparts. To visualize the differences, we have zoomed into the north-west extent of the basin as indicated in subfigures
(b). The north-west extent of the basin encompasses the East River watershed which is an area of sustained research activity
(Hubbard et al., 2018). We note the prevalence of smoother spatial gradients of at downscaled resolution (400 m) when
compared with the original resolution (800 m).





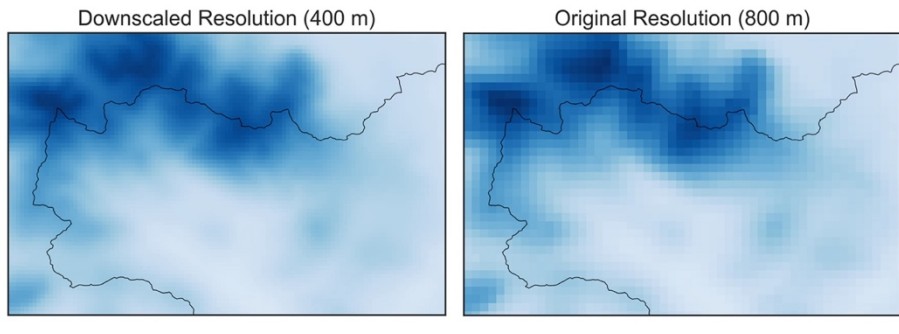

(a)

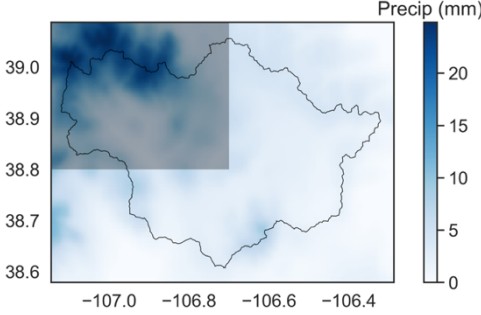

(b)

**Figure 5:** Example of downscaled precipitation grid: (a) comparison of downscaled precipitation with original precipitation (date: 5 Dec 2019), (b) gridded precipitation for the entire East-Taylor subbasin for the date in (a). The translucent box on the top left of (b) corresponds to the spatial extent shown in (a). The term 'Precip' in the colorbar is an abbreviation for precipitation.

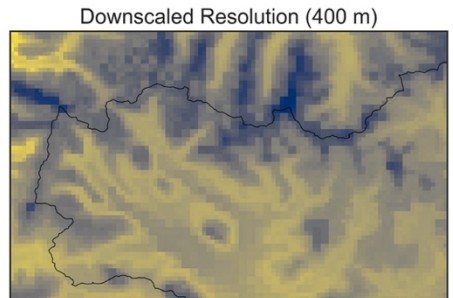

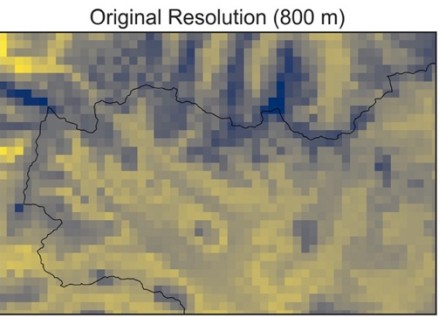

(a)



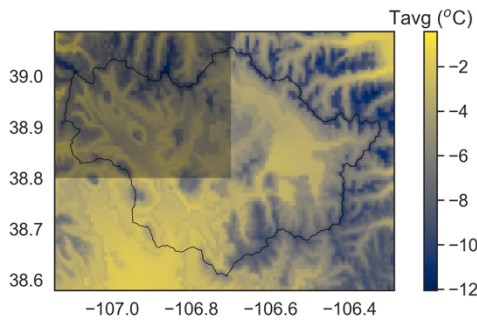

**(b)**

**Figure 6: Example of downscaled mean temperature grid: (a) comparison of downscaled temperature with original temperature (date: 5 Dec 2019), (b) gridded mean temperature for the entire East-Taylor subbasin for the date in (a). The translucent box on the top left of (b) corresponds to the spatial extent shown in (a). The term 'Tavg' in the colorbar is an abbreviation for mean temperature.**

We now quantify the smoothness (or roughness) of precipitation and temperature fields (as described in Sect. 4.1). Figure 7 shows an example of the Laplacian of precipitation at both the downscaled and original resolution, corresponding to the date shown in Fig. 5. For consistency, the precipitation at the original resolution has been projected to the downscaled grid. We note that the Laplacian for the downscaled precipitation varies smoothly. The Laplacian for precipitation at the original resolution is characterized by a checker-board pattern, which visualizes the need for generating hyper-resolution datasets

while implementing hydrological models at hyper-resolutions.

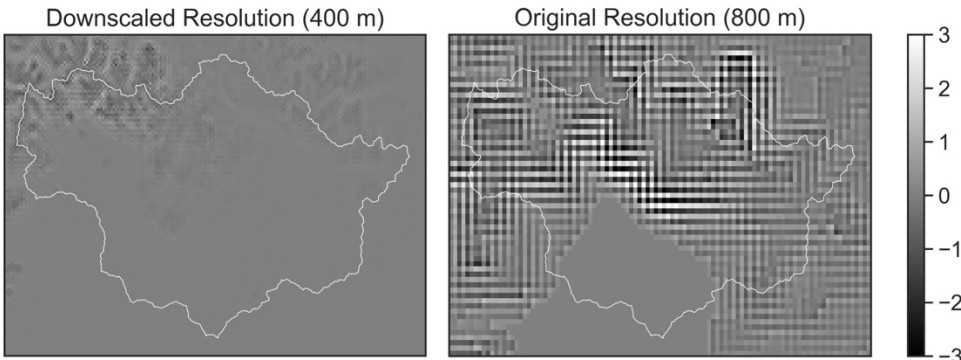

**Figure 7: Example of Laplacian of precipitation grid at the downscaled and original resolution for the date corresponding to Fig. 5. The colorbar shows the Laplacian values (units of mm/m$^2$).**

Figure 8 shows the distributions of *RR* for both precipitation and mean temperature. In both cases, the values of *RR* are well below 1, signifying that the spatial gradients of datasets are smoother at the downscaled resolution.





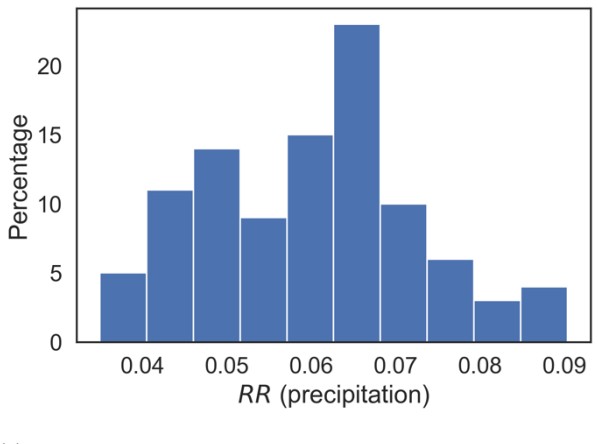

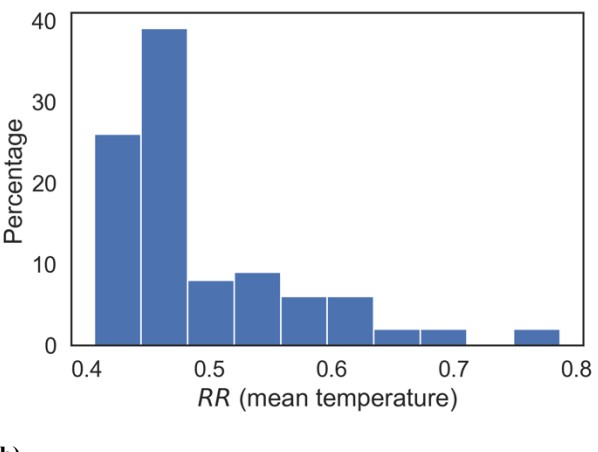

(a)                                                                    (b)

**Figure 8: Roughness ratio (*RR*) estimates for (a) precipitation and (b) mean temperature. *RR* is the ratio of downscaled roughness to original roughness shown in Eq.** Error! Reference source not found.**, where *RR* < 1 implies that the datasets are smoother at the downscaled resolution.**

**5.2    Residual error and spatial consistency of downscaled datasets**

Figure 9 shows the distributions of mean residue, where each instance corresponds to the mean residual error for a randomly selected time point. We observe that the estimated values of mean residue (as indicated by the peak value of the histogram) for both precipitation and mean temperature are close to zero. The small amount of residual error can be attributed to the fact that the downscaled dataset is generated using a machine learning model $\bar{f}$, which is an approximation of the true function $f$.

Mean residual values of zero imply that the downscaled estimates are spatially consistent with the dataset at the original resolution.

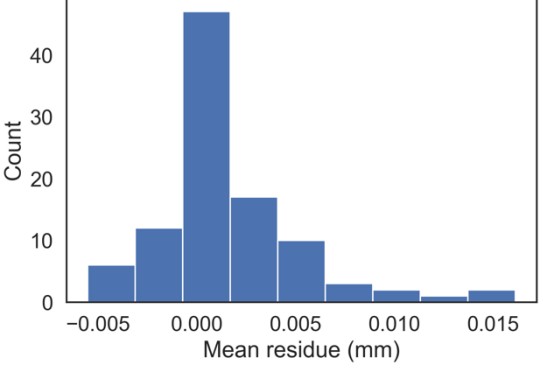

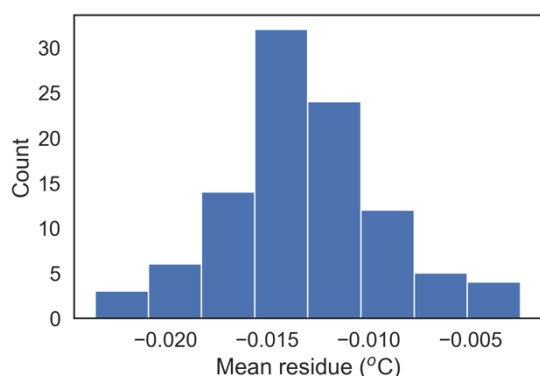

(a) Precipitation                                        (b) Mean temperature

**Figure 9: Spatial consistency check results for (a) precipitation, and (b) mean temperature. The estimated value of mean residue for precipitation is 0.002 mm, with a standard error of 0.004 mm. The estimated value of mean residue**





**for mean temperature is -0.01 ºC, with a standard error of 0.004 ºC.**

### 5.3 Effect of elevation on datasets

Figure 10 shows PDPs that marginalize the effect of elevation on each climate variable. For both climate variables, we show PDPs for 10 randomly selected days. Although PDPs were obtained for 100 days (as documented in the beginning of Sect. 5), we show results only for 10 days to prevent overcrowding of the plots. Furthermore, to enable visualization of multiple partial dependence functions on the same plot, we shifted each function by its mean value. The machine learning model used to generate downscaled datasets captures an increase (decrease) in precipitation (mean temperature) with increase in
elevation, which is consistent with the local climatology and the PRISM datasets (Daly et al., 2008).

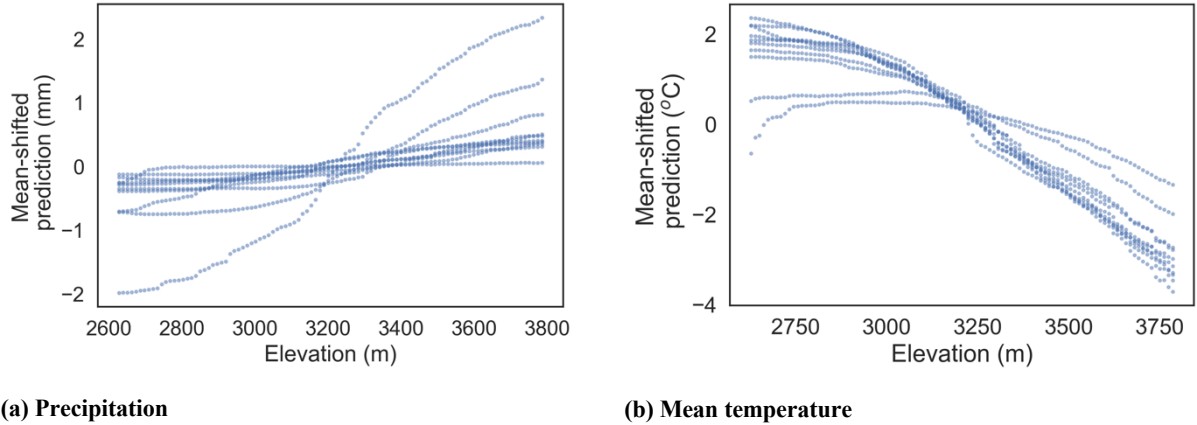

(a) Precipitation                                        (b) Mean temperature

**Figure 10: Climate-elevation relationship visualized using PDP for (a) precipitation, and (b) mean temperature**

### 6   Example use case and validation: modeling snowpack estimates

The downscaled datasets cannot be validated directly since the ground truth climate field is not known. Instead, we present a
use case to demonstrate that the downscaled datasets can be effective for ecohydrological modeling in complex mountainous terrains where climate gradients can change at fine spatial scales. We developed a novel data-driven approach that models high-resolution SWE data obtained via lidar (as described in Sect. 2.2.4). Our feature space comprised of several meteorological variables at downscaled (~400m) and original (~800m) resolutions. As part of the use case, we also evaluated if using downscaled meteorological variables can improve the modeling of SWE, when compared to using meteorological
variables at original resolutions. Any improvement in snowpack (SWE) modeling can be considered as a validation of the

spatial patterns represented by the downscaled datasets. We provide a brief description of the four meteorological variables derived for this purpose, following Mital et al. (2022):

    i.    Accumulated snowfall: Snowfall is the primary mechanism behind snow accumulation. As snow accumulation takes place over the entire snow season, we consider accumulated snowfall from the start of the snow season (defined as October 1) till the date of observation of the snowpack. Precipitation on a given day is considered to be snow if the mean air temperature is less than or equal to 0°C.

    ii.    Positive degree-day sum (PDD sum): PDD sum is used to approximate the process of snowmelt and is defined as the sum of mean daily temperatures above 0°C in a given time period. We consider PDD sum from the start of the snow melt season (defined as March 15).

    iii.    Accumulated precipitation: Since snowfall is extracted from precipitation using an approximate methodology, we also consider accumulated precipitation over the entire snow season.

    iv.    Mean seasonal air temperature ($T_{mean}$): $T_{mean}$ is computed by averaging the mean daily temperatures from the start of the snow season (October 1) till the date of observation of the snowpack. This helps consider the spatial heterogeneity of temperature across a basin.

In addition, we also considered the following five topographic variables (Mital et al., 2022): (v) elevation (vi) slope, (vii) aspect, (viii) latitude, and (ix) longitude.

Figure 11 shows the schematic of the modeling approach. We developed two RF models. RF model 1 used meteorological variables (i-iv) at the original (~800 m) resolution, while RF model 2 used meteorological variables at the downscaled (~400 m) resolution. Both models use topographic variables (v-ix) at the downscaled resolution. The target variable in both cases was SWE, also at the downscaled resolution. This enabled us to isolate the effect of using downscaled estimates of meteorological variables. The hyperparameter values for the RF models are specified in Appendix B.

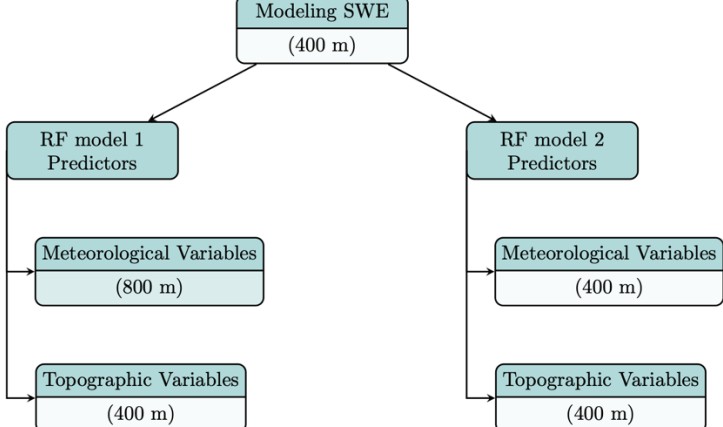

**Figure 11: Feature space for the two RF models considered in our validation exercise. Meteorological variables refer to (i) accumulated snowfall, (ii) PDD sum, (iii) accumulated precipitation, and (iv) $T_{mean}$. Topographic variables refer to**



**(v) elevation (vi) slope, (vii) aspect, (viii) latitude, and (ix) longitude.**

The RF models of spatially distributed SWE were evaluated using a leave-one-out approach. As described in Sect. 2.2.4,
there are a total of eight distinct maps available within the study area. We trained each RF model using seven maps, and
evaluated their respective abilities to model the held-out map. This exercise was conducted eight times, where each time a
different map was considered as a held-out map.  We evaluated the model performance by computing the Nash-Sutcliffe
Efficiency (NSE; Nash and Sutcliffe, 1970) on the held-out map. *NSE* is defined as:

$$NSE = 1 - \frac{MSE}{\sigma_o^2}$$
(5)


where *MSE* is the mean-squared error of the model and $\sigma_o$ is the standard deviation of the observations in the held-out map.
*NSE* is dimensionless, ranging from $-\infty$ to 1. Higher values are desirable and are consistent with lower values of *MSE*.

Table 1 shows the results of the modeling exercise, wherein we modeled SWE at 400 m resolution using both the
original (RF model 1) and downscaled (RF model 2) meteorological variables. We observed that downscaled variables yield
improvements (as indicated by higher *NSE* values) in six out of eight instances. The negative *NSE* values for *GE: 24 May
2018* are due to that particular snapshot having the lowest fractional snow cover area compared to other snapshots (Fig. 2).
This implies that the relationships between the predictors and SWE are different when compared to other snapshots (Mital et
al., 2022). Overall, the results in Table 1 suggest that even if the downscaled dataset may not capture all the spatial
variability at hyper-resolutions, it still constitutes a superior product compared to the original dataset especially when it
comes to modeling hydrological variables at hyper-resolutions. Figure 12 shows scatterplots between the observed and
predicted (modeled) SWE using RF model 2.

**Table 1: Validation results of modeling SWE at 400 m. The resolution in parenthesis refers to the resolution of
meteorological variables. Both models used topographic variables at a resolution of 400 m. For each snapshot, higher
*NSE* values are marked in bold.**

| Basin | Date | *NSE* for RF model 1 (800 m) | *NSE* for RF model 2 (400 m) |
|---|---|---|---|
| Gunnison – East River (GE) | 31 Mar 2018 | 0.61 | **0.63** |
| | 24 May 2018 | -0.74 | **-0.63** |
| | 07 Apr 2019 | 0.30 | **0.33** |
| | 10 Jun 2019 | 0.74 | 0.74 |
| Crested Butte (CB) | 04 Apr 2016 | 0.70 | **0.74** |
| Gunnison – Taylor River | 30 Mar 2019 | 0.35 | **0.44** |



| (GT) | 08 Apr 2019 | **0.62** | 0.58 |
| | 09 Jun 2019 | 0.68 | **0.69** |

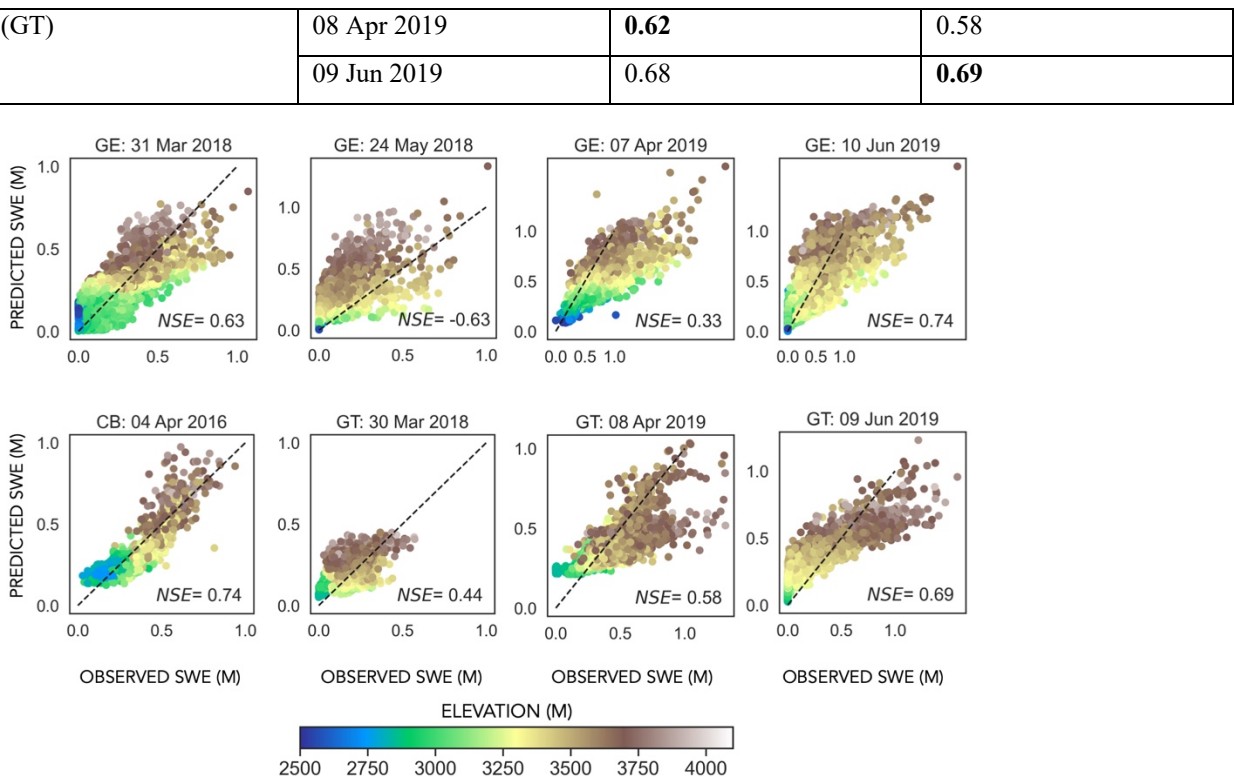

**Figure 12: Example use case of downscaled datasets showing scatterplots to predict SWE using RF model 2. The individual points are color-coded by elevation.**

## 7    Caveats and future development

The use case presented in this study points to some potential limitations of the dataset. First, the use case was designed to evaluate the aggregate effect of climate variables in mountainous regions during the snow season. This means additional use cases may need to be developed to ascertain the quality of the dataset during dry periods and summer months. Furthermore, the current use-case models snowpack using a data-driven framework that may not capture all the factors driving spatial variability of snowpack. A more detailed use case can be envisioned which may involve using downscaled meteorological datasets as forcing variables for hydrological and land surface models. Recent studies have explored how the spatial resolution of gridded meteorological variables can affect hydrological responses such as surface runoff, soil moisture and evapotranspiration (Shuai et al., 2021; Maina et al., 2020).

It is important to note that a lack of hyper-resolution observations makes it challenging to estimate the true errors associated with gridded datasets (Daly, 2006). Nevertheless, it is important to pursue development of hyper-resolution datasets (such as the ones presented in this study) so that they can be visualized and critically evaluated (Beven et al., 2015).





This will enable identification of any missing features and drive further refinement of methodologies for generating hyper-resolution datasets.

## 8    Data availability

The presented dataset is freely available on the United States Department of Energy's Environmental System Science Data Infrastructure for a Virtual Ecosystem (ESS-DIVE) repository. It can be accessed at https://doi.org/10.15485/1822259 and

cited as Mital et al. (2021). We recommend accessing the dataset using Chrome or Firefox browsers. The dataset consists of two zip files: one for daily precipitation and one for daily mean temperature. The data are arranged by year and are in the NetCDF format, which is a standard raster format that can be read using Geographic Information System software and popular scripting languages (e.g, R, Python, MATLAB). The datasets have been projected to the coordinate system denoted by NAD83 / UTM zone 13N – EPSG:26913. A Jupyter notebook has been provided as a supplement which illustrates the

spatial downscaling methodology.

## 9    Conclusions

We have presented a description of a hyper-resolution (400 m) gridded dataset of daily precipitation and mean temperature. The datasets cover a 12-year period of 2008-2019. The spatial extent of the datasets is the East-Taylor subbasin, which is a mountainous watershed and is an important study area in the context of water security for southwestern United States and

Mexico (Sect. 2.1). The datasets were generated by downscaling daily gridded datasets developed by the PRISM group (800 m resolution). Rather than seeking to train on paired coarse-resolution and fine-resolution data (which are not available), our methodology sought to learn relationships between topographic features and daily climate variables. These relationships were constrained or regularized by the use of nearest neighbor maps that forced the machine learning model to more explicitly consider point observations. The relationships were then implemented to generate downscaled datasets.

Downscaling enabled us to leverage knowledge about physiographic factors and climatological processes that are embedded in the existing datasets.

The precipitation and temperature fields at the downscaled resolution provide a more precise definition of local gradients when compared with the original dataset. This can aid in the implementation of hydrological and land surface models in complex mountainous terrains with fine-scale spatial gradients. The downscaled fields are spatially consistent with

the original dataset with a mean residual error that is approximately zero. We observe the prevalence of linear relationships between climate variables and elevation, which is consistent with the PRISM datasets. Finally, we demonstrated a use case for downscaled datasets by implementing a data-driven framework to model snowpack. The presented dataset constitutes a valuable resource to implement ecohydrological and land surface models in the mountainous terrain of the East-Taylor subbasin.



## Appendix A: Additional quality control for lidar observations

The lidar maps corresponding to Gunnison-East River (GE) required additional quality control measures. First, a number of pixels in the GE maps appeared to be numerical artifacts. To remove these artifacts, we assumed that the map labeled *GE: 31 Mar 2018* recorded a continuous snow cover (given that the date is close to peak SWE; Clow, 2010) Therefore, any pixels with SWE ≤ 0 were masked. The unmasked pixels gave us an initial spatial extent for GE maps. However, this initial spatial extent exceeded the spatial extent for the map labeled *GE: 10 Jun 2019*. Therefore, we considered an intersection of the two spatial extents, which yielded a consistent spatial extent across all four GE maps. This spatial extent is shown in Fig. 2(b). Subsequently, we cropped part of the GE maps that fell outside the East-Taylor subbasin, yielding a final spatial extent as shown in Fig. 2(a).

## Appendix B: Hyperparameters for RF models

We used RF to learn the function $f$ for spatial downscaling, as well for modeling snowpack in our example use case. Concerning the choice of hyperparameters for both sets of RF models, we sought to use values that were specified as default choices by the developers of RF (as documented at https://CRAN.R-project.org/package=randomForest). Therefore, we specified 500 trees, considered $p/3$ features when looking for the best split (where $p$ is the number of predictor variables) and specified a node size (i.e., minimum number of samples in a leaf node) of 5. For spatial downscaling, we specified a node size of 1 (instead of 5) to encourage growth of deep trees. The use of nearest neighbors (i.e, $w_{1-10}$) as predictors helped alleviate any overfitting that may happen with deep trees. Note that the number of predictor variables (i.e, $p$) is 13 for spatial downscaling, and 9 for the use case.

## Appendix C: Feature importance of RF models

Figure C1 shows estimates of feature importance of RF models trained to downscale precipitation and mean temperature. For precipitation, we observe that all three topographic variables (i.e., elevation, longitude, latitude) along with the first two nearest neighbors (i.e., $w_1$ and $w_2$) are important for model predictions. For mean temperature, elevation has an outsized influence, with nearest neighbors having very little impact on the model predictions. Daily precipitation exhibits high spatial variability which is not necessarily a function of changes in local topography (e.g., rain shadow effect, seeder-feeder mechanisms) making it important to explicitly consider information from nearby weather stations. The variability of mean daily temperature is more closely related to changes in elevation on account of the adiabatic lapse rate. As a result, there is a lower dependence on other factors.

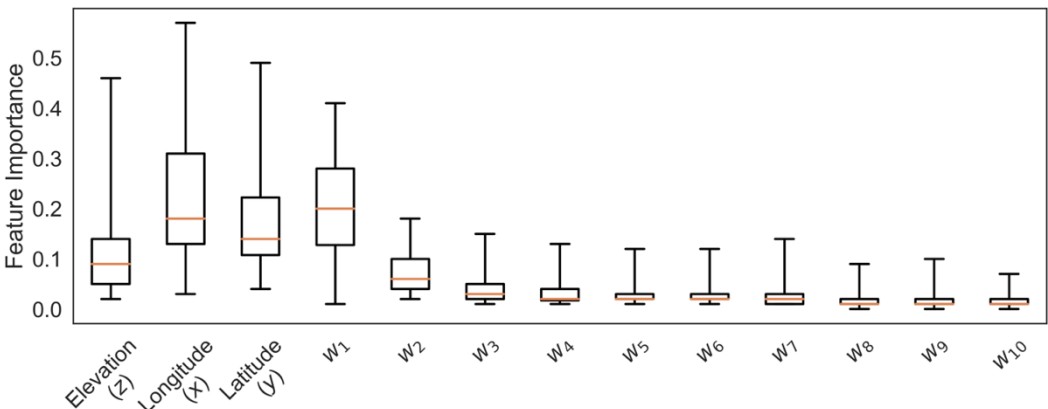

**(a) Precipitation**

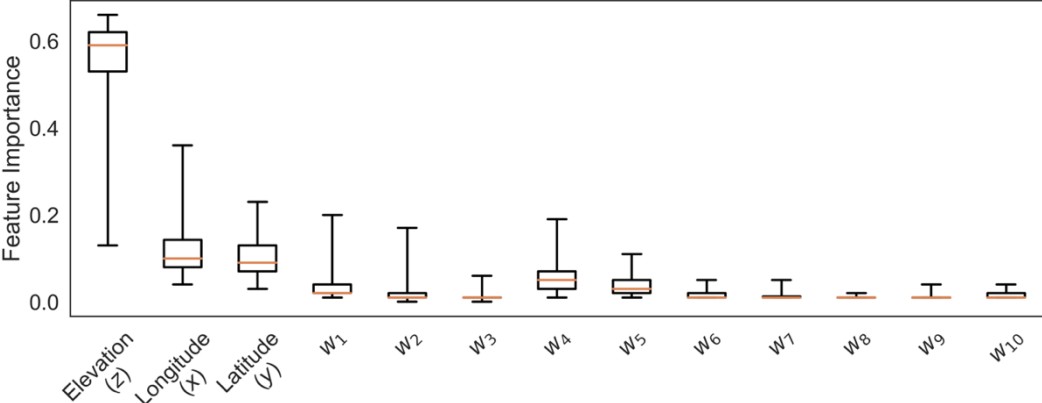

**(b) Mean temperature**

**Figure C1: Feature importance box plots for (a) precipitation, and (b) mean temperature**

## Author contribution

UM and DD conceived the study. UM curated the data, developed and implemented the downscaling methodology, and verified and validated the generated datasets. DD and JB provided input on the overall methodology. CS provided input on the conception of the study, provided overall supervision and acquired funding for the study. UM prepared the manuscript with contributions from all co-authors.

## Competing Interests

The authors declare that they have no conflict of interest.



**Acknowledgements**

This work was funded by the ExaSheds project, which was supported by the U.S. Department of Energy, Office of Science, Office of Biological and Environmental Research, Earth and Environmental Systems Sciences Division, Data Management Program, under Award Number DE-AC02-05CH11231. The proprietary PRISM data (800 m resolution) was purchased with funding from the Watershed Function Scientific Focus Area funded by the U.S. Department of Energy, Office of Science,

Office of Biological and Environmental Research under Award no. DE-AC02-05CH11231.

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
