# Peer review of "Downscaled hyper-resolution (400 m) gridded datasets of daily precipitation and temperature (2008-2019) for East Taylor subbasin (western United States)"

_Earth System Science Data, 2022_

## Author Comment (AC1)

**Reviewer 1**

The paper presents a methodology to downscale precipitation data using topography as a covariate. The paper is technically sound and it presents a pragmatic and robust methodology for downscaling.

Reply: Thank you for the positive feedback on our study.

**Reviewer 2**

The manuscript by Mital et al. attempts to downscale PRISM data from 800 to 400m spatial resolution in the East-Taylor subbasin. The topic is of great interest to ecohydrologists. The presentation is very clear. However, there are many questions that need to be addressed by the Authors:

Reply: Thank you for the review. We have addressed the questions below. References to works that are cited here can be found at the end of the response document.

[1] PRISM data is available for the United States (https://climatedataguide.ucar.edu/climate-data/prism-high-resolution-spatial-climate-data-united-states-maxmin-temp-dewpoint). If I understood it correctly, the only fine-scale information used in the downscaling approach is related to topography. Then the obvious question is why not extend it to the larger area because 10m DEM is available from NED. I acknowledge that the importance of the selected study area is clearly mentioned in Section 2.1. But similar reasonings (as in Section 2.1) can be presented for some other areas as well.

Reply: We clarify that in addition to topography, we also use nearest neighbor maps that are developed using weather station data. As discussed in section 2.3.2, and further elaborated in response to comment [4], weather station data needs to be subjected to two steps of pre-processing. Therefore, while 10m DEM is available throughout the United States, nearest neighbor maps (that rely on processed weather station data) are not. As a result, extending the study area to a larger area requires substantial effort.

Additionally, the chosen study area is co-located with several watersheds where research on water availability and quality is ongoing. We have updated Figure 1 in the manuscript to highlight some such study areas. The updated figure is reproduced below.

[Figure]

Figure 1. Location of East-Taylor subbasin in the western United States. Also shown are several watersheds within East-Taylor that are subjected to research on water availability and quality.

This is accompanied by the following new text in the first paragraph of section 2.1:
*"East-Taylor subbasin encompasses several watersheds including East River, Taylor River and Coal Creek. These watersheds have been subjected to intensive research activity,..."*

Research in the above study areas is funded by the US Department of Energy (which also funded the current study as specified under "Acknowledgments"). This was an additional motivation for prioritizing the selected area for the current study.

Finally, we point out that generating datasets at hyper-resolution necessitated the development of a new methodology (as discussed in section 1). Therefore, we were also motivated by the need to evaluate the generated dataset for a smaller study area before expanding to a larger area.  We have added the following additional clarification at the end of section 2.1:
*We start with a relatively small study area within UCRB to make it easier to visualize and critically evaluate the dataset. The novelty of the downscaling methodology motivates us to build confidence with a smaller study area (i.e., East-Taylor subbasin) before expending resources to generate datasets that cover a larger area (i.e., UCRB and beyond). Since the East-Taylor subbasin is an area of intensive research activity, the generated datasets can be rapidly incorporated in land-surface and ecohydrological modeling. This will also help us to identify any missing features in the datasets which may drive further refinement of the underlying downscaling methodology (Sect. 3).*

We have also added the following statement in section 7 (Caveats and future development):
*"Future work will extend the study area to a larger extent (i.e., UCRB and beyond)."*

[2] If the topography data is available at 10m and the method heavily depend on the topography, then why Authors chose 400 m as grid size for the downscaling?
Reply: As clarified in response to comment [1] above, the method also depends on nearest neighbor maps. Nevertheless, we had still experimented with finer grid sizes (200m and 100m). We observed that in our study area, which is a mountainous watershed, the quality of the downscaled dataset tended to deteriorate at finer scales. This could be because at smaller scales, the spatial variability of meteorological variables in mountainous regions is controlled by fluxes that are not prevalent at coarser scales (800 m and up). For instance, at spatial resolutions finer than 100m, spatial variability of mountainous precipitation is affected by small-scale processes such as particle-flow interaction and riming of snow particles (Mott et al. 2018). Widespread data that observes such small-scale processes is typically not available.

The current study used topography and nearest neighbor maps to generate downscaled datasets. It is possible that additional factors such as soil, geology and vegetation also become important (Beven et al. 2015) and may need to be considered while downscaling to finer resolutions.

We have added the following clarification to the penultimate paragraph of section 1 (new text is italicized):
"This approach has the benefit of leveraging expert knowledge about physiographic factors and climatological processes that is embedded in the gridded datasets. *However, it is limited in its ability to introduce new knowledge about physical processes at smaller scales (< 800 m). For instance, it is challenging to account for the effect of small-scale processes (100 m or less) on precipitation such as particle-flow interaction and snow riming (Mott et al., 2018). Therefore, we set the scope of the current study to downscaled datasets at a resolution of 400 m.*"

Additional research is needed to reliably downscale to finer resolutions. We have added the following clarification at the end of section 7 (Caveats and future directions):
*"Additional research is needed to reliably downscale gridded datasets to finer resolutions (i.e., beyond 400 m), and it may require us to consider additional information (e.g., soil cover, canopy, multispectral satellite data, radar data)."*

[3] I assume that by precipitation map Authors mean both the snow and rainfall, is it correct? Looking at Figure 2, it is unclear how the SWE maps relate to hyper-resolution precipitation and temperature.
Reply: Yes, by precipitation map, we indeed mean both rain and snow. We have added a clarification in section 2.2.1 while describing the data source (new text is italicized):
"We obtained gridded estimates of daily precipitation *(which includes both rain and snow)...*"

Concerning Figure 2, the SWE maps do not directly relate to hyper-resolution precipitation and temperature. We specify in section 2.2.4 that the SWE maps constitute independent datasets that are used exclusively to demonstrate an example use case of the downscaled datasets, and are not used in the downscaling methodology itself. The relationship of SWE maps to hyper-resolution precipitation and temperature is addressed in the use case in section 6.1.

We have added the following clarification to the caption of Figure 2:
*"These maps constitute independent datasets that are not used for downscaling precipitation and temperature, but for demonstrating an example use case (Sect. 6.1)."*

[4] Authors have referred to their previous work in Section 2.3.2 while mentioning about the gap filling in the weather data. This is a very crucial step. The justification is not very clear about using a data-driven imputation approach.

Reply: We acknowledge the reviewer's concern. The benefits of the data-driven sequential imputation approach were demonstrated in our previous works (Mital et al., 2020; Dwivedi et al., 2022). To clarify the justification for using a data-driven sequential imputation approach, we have added the following lines at the end of section 2.3.2 (*new text is italicized*):

*"We gap-filled missing values of precipitation and temperature using a data-driven sequential imputation approach (Mital et al., 2020; Dwivedi et al., 2022). This approach helps to gap-fill missing values using neighboring weather stations. Importantly, this approach overcomes two key limitations of other imputation approaches, in that they do not require (i) specification of a functional form to do a weighted interpolation using neighboring weather stations, and (ii) neighboring weather stations to have a complete time series. In particular, we used the approach detailed in Mital et al. (2020) which was developed specifically for meteorological variables."*

[5] Figure 4 shows the nearest neighbour map. It is not clear whether correlations were calculated based on the values at the grids or at the centroid of the polygons. Authors have rightly mentioned that the nearest neighbor for a grid point is not necessarily the closest station, especially in the area with different topography. Then what it means to have correlated stations, it is unclear.

Reply: Correlations are calculated based on the values at individual grid points. We explain in section 3.2 that we compute correlations between each grid point and the weather stations. This helps us determine which stations are the most correlated (or are the nearest neighbors) for each grid point. In the nearest neighbor map shown in Figure 4, each grid point is color-coded using the color of its first nearest neighbor (or w_1). Consequently, the map ends up looking like a collection of polygons, where each polygon comprises grid points that have the same first nearest neighbor. However, there is no special meaning attributed to the shapes or centroids of these polygons.

To improve clarity, we have updated the caption of Figure 4 to read "first nearest neighbor" instead of just "nearest neighbor" and have added the following sentence (new text in italics):

Each grid point is color-coded using the color of its "first nearest neighbor" weather station. *Consequently, the map resembles a collection of polygons where each polygon comprises grid points that have the same first nearest neighbor.*

[6] In the Section 4.2, Authors mention that 'if the downscaled datasets are aggregated to the original resolution, we should get back the original dataset'. This means Authors are trying for disaggregation, not spatial downscaling. Both are not the same. Authors

may need to apply IDW type of approach to estimate the value at the centroid of the larger grid using the values at the smaller grids within the larger grid.

Reply: We clarify that our intent behind quantifying residual error is to ensure that there is no bias present in the downscaled dataset. We understand that the quoted sentence does not express this properly and can give the impression that we are trying for disaggregation. Therefore, we have modified the sentence as follows:

*"The process of downscaling should not introduce any bias in the hyper-resolution datasets. We can verify this by upscaling the downscaled datasets back to the original resolution (i.e., 800 m) and quantifying the mean residual error with respect to the original dataset (also at 800 m resolution)."*

Additionally, we clarify that when we refer to aggregation, we infact mean upscaling which is achieved via bilinear interpolation. We point out that when upscaling by a factor of 2, the equations of IDW and bilinear interpolation reduce to taking the mean of (or aggregating) a 2x2 grid. To avoid confusion, we have substituted the word "aggregated" with the word "upscaled". We have also added the following clarification to section 4.3 (which was section 4.2 in the original submission):

*"The upscaling was done via bilinear interpolation."*

[7] In the beginning of section 5, the wet days are defined as precipitation>1mm. There has to be some justification for this. Why not chose 0.01 or 0.5 mm?

Reply: 1 mm corresponds to the resolution of the weather station data. We have added the following clarification (in italics) where we define wet days:

*"…wet days (when mean precipitation across East Taylor was greater than 1 mm, which corresponds to the resolution of the weather station data)."*

[8] For the downscaled precipitation, only one figure (Figure 5) is presented to visually match the spatial patterns. Authors should look into spatial statistics such as Variogram, skill scores etc. to objectively match the downscaled value to the reference.

Reply: Thank you for this suggestion. We have added a new subsection in section 4 entitled "Quantifying spatial variability". This section reviews the concept of a variogram and presents the equation used for computing an empirical semi-variogram. Subsequently, we compute semi-variograms for both precipitation and temperature fields. We present the following example variograms in section 5.2 (see Figure 9 in the revised manuscript):

[Figure]

These variograms correspond to the date in Figures 5 and 6 (i.e., 5 Dec 2019). Indeed, we observe that the spatial patterns of the downscaled values objectively match the reference values at the original resolution. We make the following statement in section 5.2:

*"The variability at the downscaled resolution is similar to that at the original resolution. This shows that the downscaled datasets preserve the spatial structure of the climate field present in the original datasets."*

[9] Section 5.2 is very confusing. It is unclear why the mean residual error is calculated temporally when Authors are talking about spatial downscaling. Please consider expanding lines 280 to 285 as these are very crucial information to justify the downscaled estimates.

Reply: We acknowledge the confusion. The mean residual is in fact calculated spatially (by upscaling the downscaled dataset and comparing with the original dataset), as described in section 4.3 (which was 4.2 in the original submission). We have added the following clarification in section 5.3 (which was 5.2 in the original submission):

*We clarify that "mean" in this context refers to the spatial mean over the entire study area.*

This is accompanied by a similar clarification that has been added to section 4.3:

*"For a given time point, we quantify the mean residual error over the entire study area …"*

We get a value of the spatial mean residue for each time point. We then consider several randomly selected time points to get a probability distribution of the spatial mean residue. This helps to visualize the spread. We have clarified this in the revision at the start of section 5:

*"While estimating roughness, mean residual error, and partial dependence (outlined in Eq. (2), (4) and (5), respectively), we seek to visualize their spread rather than obtaining a single value. Therefore, we randomly sampled 100 time points (or days) and computed the above error metrics for each of those time points."*

Additionally, we have also added a reference to section 4.3 at the start of section 5.3 (which was 5.2 in the original submission). This helps to remind the reader why the mean residual error is calculated in the first place.

[10] In Section 6, the example use case is of ecohydrological modeling, more specifically in terms of modeling of SWE. Since there is not much mentioned about the methodology in the manuscript, it looks surprising why Authors couldn't use any hydrological models to check the streamflow in the mountainous area, why specifically SWE modeling? The discussion so far was about the downscaled Precipitation and Temperature, which itself has uncertainty. Now Authors have estimated four new meteorological variables and set up two RF models (Figure 11). I assume that all these are done at every grid, then how to take into account the spatial structure of precipitation and temperature.

Reply: Thank you for this comment. A detailed description of the SWE modeling methodology can be found in our recently published work (Mital et al., 2022), which was still under review at the time of the original submission. This work has been cited in the manuscript while referring to Figure 12 (which was Figure 11 in the original submission):

*"Figure 12 shows the schematic of the modeling approach, based on our previous work (Mital et al., 2022)."*

Additionally, the use of hydrological models to check the streamflow (and several other hydrological response variables) in mountainous areas was recently published as part of a separate study (Shuai et al., 2022). We have added a brief review of the study in section 6 as follows:

**6.2 Additional use cases: impact of meteorological forcing resolution on hydrological responses**

*Additional use cases of downscaled datasets involve studies that investigate the impact of spatial and temporal resolution of gridded meteorological forcing on watershed hydrological responses (Shuai et al., 2022; Maina et al., 2020). For instance, Shuai et al. (2022) explored the effects of spatial and temporal resolution of gridded meteorological forcing on watershed hydrological responses. The study used integrated hydrological modeling and was conducted in the Coal Creek watershed, which is a mountainous sub-watershed located at the western edge of the East-Taylor subbasin. The downscaled daily datasets were used as high-resolution forcing variables, and the simulated streamflow was found to be consistent when compared with the results of coarser resolution forcings. The study also considered a number of additional hydrological variables (i.e., SWE, snowmelt, ponded depth, groundwater level, soil moisture and evapotranspiration). For more details, we refer the reader to Shuai et al. (2022).*

The original use case presented in our study focuses on modeling SWE because it is a spatially distributed variable (as opposed to streamflow). Spatially distributed hydrological responses are more likely to be affected by the spatial resolution of meteorological forcings. The four new meteorological variables considered in the use case are temporal aggregations of precipitation and temperature, which preserves the overall spatial structure. We have added the following clarification in section 6.1 immediately after describing the four variables:

*"Note that the above variables are temporal aggregations, and as such the spatial structure of downscaled precipitation and temperature is preserved."*

[11] The title of the manuscript doesn't mention that the Authors have checked the applicability of the downscaled dataset only for the snow season. In Section 7, it is clearly mentioned that for dry periods and summer months, the Authors are not sure about the quality of the dataset. Then I suggest changing the title of the manuscript.
Reply: The datasets have been developed and made available for all seasons. Changing the title of the manuscript would imply otherwise. While a use case specific to Summer months has not been presented, the verification checks in section 5 are independent of the season. We also point out that the use case does cover the start of the Summer months (i.e., early June). Additionally, the recently published study by Shuai et al. (2022), a brief review of which was added to the manuscript in response to comment [10] above, models hydrological response of a watershed for the entire year. This provides more confidence in the quality of the dataset. Therefore, we have also deleted the statement about uncertainty regarding dry periods and summer months. However, more robust evaluations of the downscaled datasets may still be needed, and we continue to acknowledge those caveats in section 7. We have updated a discussion of the caveats as follows:

*"The use cases presented and reviewed in this study evaluated the impact of using downscaled meteorological variables for modeling the hydrological response in mountainous regions. As the presented use-case modeled snowpack using a data-driven framework, it is possible that not all the factors driving spatial variability of snowpack were considered. Additional evaluation of downscaled datasets may require access to spatially distributed ground-truth data at hyper resolutions (e.g., via X-band radar), as well as comparisons with hyper-resolutions outputs (if available) of land-surface and numerical weather prediction models."*

References:

Beven, K., Cloke, H., Pappenberger, F., Lamb, R., and Hunter, N.: Hyperresolution information and hyperresolution ignorance in modelling the hydrology of the land surface, Sci. China Earth Sci., 58, 25–35, https://doi.org/10.1007/s11430-014-5003-4, 2015.

Dwivedi, D., Mital, U., Faybishenko, B., Dafflon, B., Varadharajan, C., Agarwal, D., Williams, K. H., Steefel, C., and Hubbard, S.: Imputation of Contiguous Gaps and Extremes of Subhourly Groundwater Time Series Using Random Forests, J Mach Learn Model Comput, 3, 1–22, https://doi.org/10.1615/JMachLearnModelComput.2021038774, 2022.

Maina, F. Z., Siirila-Woodburn, E. R., and Vahmani, P.: Sensitivity of meteorological-forcing resolution on hydrologic variables, Hydrol. Earth Syst. Sci., 24, 3451–3474, https://doi.org/10.5194/hess-24-3451-2020, 2020.

Mital, U., Dwivedi, D., Brown, J. B., Faybishenko, B., Painter, S. L., and Steefel, C. I.: Sequential Imputation of Missing Spatio-Temporal Precipitation Data Using Random Forests, Front. Water, 2, 20, https://doi.org/10.3389/frwa.2020.00020, 2020.

Mital, U., Dwivedi, D., Özgen-Xian, I., Brown, J. B., and Steefel, C. I.: Modeling Spatial Distribution of Snow Water Equivalent by Combining Meteorological and Satellite Data with Lidar Maps, Artificial Intelligence for the Earth Systems, in press, https://doi.org/10.1175/AIES-D-22-0010.1, 2022.

Mott, R., Vionnet, V., and Grünewald, T.: The Seasonal Snow Cover Dynamics: Review on Wind-Driven Coupling Processes, Front. Earth Sci., 6, https://doi.org/10.3389/feart.2018.00197, 2018.

Shuai, P., Chen, X., Mital, U., Coon, E. T., and Dwivedi, D.: The effects of spatial and temporal resolution of gridded meteorological forcing on watershed hydrological responses, Hydrol. Earth Syst. Sci., 26, 2245–2276, https://doi.org/10.5194/hess-26-2245-2022, 2022.